# Interactions between Autophagy, Proinflammatory Cytokines, and Apoptosis in Neuropathic Pain: Granulocyte Colony Stimulating Factor as a Multipotent Therapy in Rats with Chronic Constriction Injury

**DOI:** 10.3390/biomedicines9050542

**Published:** 2021-05-12

**Authors:** Ming-Feng Liao, Shin-Rung Yeh, Kwok-Tung Lu, Jung-Lung Hsu, Po-Kuan Chao, Hui-Ching Hsu, Chi-Hao Peng, Yun-Lin Lee, Yu-Hui Hung, Long-Sun Ro

**Affiliations:** 1Department of Neurology, Chang Gung Memorial Hospital, College of Medicine, Linkou Medical Center and Chang Gung University, Taipei 33305, Taiwan; mingfengliao@hotmail.com (M.-F.L.); tulu@ms36.hinet.net (J.-L.H.); esther0227@gmail.com (Y.-L.L.); namiado@gmail.com (Y.-H.H.); 2Department of Life Science, National Taiwan Normal University, Taipei 11677, Taiwan; ktlu@ntnu.edu.tw; 3College of Life Science, National Tsing Hua University, Hsinchu 30013, Taiwan; sryeh@life.nthu.edu.tw; 4Department of Neurology, New Taipei Municipal TuCheng Hospital, Chang Gung Memorial Hospital, New Taipei City 23652, Taiwan; 5Graduate Institute of Humanities in Medicine and Research Center for Brain and Consciousness, Shuang Ho Hospital, Taipei Medical University, Taipei 23561, Taiwan; 6Institute of Biotechnology and Pharmaceutical Research, National Health Research Institutes, Miaoli 35053, Taiwan; chaopk@gmail.com; 7Division of Chinese Acupuncture and Traumatology, Chang Department of Traditional Chinese Medicine, Gung Memorial Hospital, College of Medicine, Linkou Medical Center and Chang Gung University, Taipei 33305, Taiwan; faithjanet@gmail.com (H.-C.H.); toponchu@gmail.com (C.-H.P.)

**Keywords:** neuropathic pain, granulocyte colony stimulating factor, chronic constriction injury, autophagy, apoptosis, proinflammatory cytokine

## Abstract

Our previous studies have shown that early systemic granulocyte colony-stimulating factor (G-CSF) treatment can attenuate neuropathic pain in rats with chronic constriction injury (CCI) by modulating expression of different proinflammatory cytokines, microRNAs, and proteins. Besides the modulation of inflammatory mediators’ expression, previous studies have also reported that G-CSF can modulate autophagic and apoptotic activity. Furthermore, both autophagy and apoptosis play important roles in chronic pain modulation. In this study, we evaluated the temporal interactions of autophagy, and apoptosis in the dorsal root ganglion (DRG) and injured sciatic nerve after G-CSF treatment in CCI rats. We studied the behaviors of CCI rats with or without G-CSF treatment and the various levels of autophagic, proinflammatory, and apoptotic proteins in injured sciatic nerves and DRG neurons at different time points using Western blot analysis and immunohistochemical methods. The results showed that G-CSF treatment upregulated autophagic protein expression in the early phase and suppressed apoptotic protein expression in the late phase after nerve injury. Thus, medication such as G-CSF can modulate autophagy, apoptosis, and different proinflammatory proteins in the injured sciatic nerve and DRG neurons, which have the potential to treat neuropathic pain. However, autophagy-mediated regulation of neuropathic pain is a time-dependent process. An increase in autophagic activity in the early phase before proinflammatory cytokines reach the threshold level to induce neuropathic pain can effectively alleviate further neuropathic pain development.

## 1. Introduction

Previous studies, including ours, have shown that granulocyte colony-stimulating factor (G-CSF) treatment can attenuate neuropathic pain in rats with chronic constriction injury (CCI) and spinal cord injury [1,2,3,4,5]. Human clinical trials have also demonstrated the analgesic effects of G-CSF in patients with compressive myelopathy [6,7]. Previous studies have reported that G-CSF treatment recruits opioid-containing polymorphonuclear granulocytes (PMNs), upregulates Mu opioid receptors (MORs) on injured nerves, suppresses proinflammatory cytokines (Interleukin-6 [IL-6], Tumor Necrosis Factor-α [TNF-α] and monocyte chemoattractant protein-1 [MCP-1]) in dorsal root ganglia (DRG) neurons and downregulates phosphorylated p38 and microglial activity in the spinal dorsal horn, thus attenuating neuropathic pain in CCI rats [1,2,3,4,5]. G-CSF is a type of multipotent cytokine that increases the circulating progenitor cells of multiple hematopoietic lineages and interacts with multiple cytokines [8]. In addition to modulating different cytokines, G-CSF also activates autophagy and inhibits apoptosis [9,10]. Guo et al. reported that systemic G-CSF treatment in rats with spinal cord injury could improve motor function and reduce damage to neural tissue in the spinal cord by promoting autophagy and inhibiting apoptosis [9]. Several animal studies have shown that G-CSF exhibits a neuroprotective effect through the activation of the antiapoptotic pathway in cerebral ischemic injury [10,11,12,13,14]. Ex vivo studies also showed that G-CSF could activate autophagic activity in hematopoietic stem cells (HSCs) [15] and delay neutrophil apoptotic activity [16].

Autophagy is a process that digests damaged cellular organelles and improves the survival rates of cells under stress conditions [17,18]. Autophagy plays an important role in preventing cancer, immune diseases, and neurodegenerative diseases such as Parkinson’s disease and Alzheimer’s disease [17,18,19]. Recent studies have also shown that autophagy is activated in injured nerves, DRG neurons, and the spinal dorsal horn after nerve injury and plays an important role in the modulation of neuropathic pain [20,21,22,23,24,25,26,27,28,29,30]. Different studies have shown that rapamycin, metformin, hyperbaric oxygen (HBO) therapy, and even microRNA therapy, which upregulates autophagy, can attenuate neuropathic pain [23,24,25,28,29]. In contrast, 3-methyladenine (3-MA) downregulates autophagy and enhances neuropathic pain in CCI rats [24]. Apoptosis is another cellular response to stress. Apoptosis has been reported in DRG neurons and satellite glial cells after peripheral nerve injury [31,32,33,34]. Several studies have shown that drugs, such as exogenous erythropoietin, saffron extracts, and metformin, and HBO therapy can attenuate neuropathic pain in different animal models, possibly by inhibiting apoptotic activity in the spinal cord and DRG neurons [35,36,37,38]. In this study, we aimed to investigate whether G-CSF treatment could attenuate neuropathic pain by modulating autophagic/apoptotic activity in injured nerves and DRG neurons in CCI rats. We studied the interactions between autophagy, proinflammatory cytokines, and apoptosis in DRG neurons and injured sciatic nerves in CCI rats with or without G-CSF treatment at different time points after nerve injury using behavioral tests, Western blot analysis, and immunohistochemical methods.

## 2. Materials and Methods

### 2.1. Animals

We used adult male Sprague Dawley rats (BioLASCO Taiwan Co., Ltd., Taipei, Taiwan) weighing 300–350 g. The rats were housed in a temperature-controlled (22 °C) cage; rat chow pellets and water were available ad libitum. All behavioral tests were performed during the light cycle. All procedures were conducted in a manner that minimized animal suffering and were approved by the Institutional Animal Care and Use Committee (IACUC) of the Chang Gung Memorial Hospital (number: 2017092504).

### 2.2. Surgical Procedure and G-CSF Treatment

Sodium pentobarbital (50 mg/kg body weight) was injected intraperitoneally (i.p.) to anesthetize the rats before the operation. The rats underwent CCI surgery according to Bennett’s model [39]. Muscles on the right inguinal region were separated by forceps to expose the sciatic nerve. Four 4–0 chromic gut ligatures were loosely tied around the proximal part of the right sciatic nerve with approximately 1.0–1.5 mm intervals between each knot. The ligatures barely reduced the nerve diameter and epineural circulation was preserved when the knots were tied. Muscle dissections without nerve ligatures were performed in the sham group. A single dose of G-CSF (200 μg/kg, Filgrastim; Kyowa Hakko Kirin, Japan) was injected intravenously (i.v.) immediately after surgery. An equal amount of normal saline was injected into vehicle-treated CCI rats.

### 2.3. Behavioral Tests for Mechanical Allodynia

Behavioral analyses were performed on the 1st, 3rd, and 7th days after surgery. Each animal was placed in a 30 × 30 × 15 cm transparent box. Each animal was allowed 10 min for habituation before the behavior tests. We tested mechanical allodynia using the von Frey hair test according to a previously described protocol [1,2,5]. We applied von Frey hairs to the central region of the right plantar surface of the hind paw in ascending order of force (1.4, 2.0, 4, 6, 8, 10, 15, and 26 g). When the rats showed a flinch or a sharp withdrawal response to the given filament, the bending force of that filament was defined as the mechanical threshold intensity. When a withdrawal response was established, a filament with the next lower force was used, and the test was restarted in ascending order. The hind paw withdrawal threshold was defined as the lowest force that caused at least three withdrawals from five consecutive applications [40]. The experimental conditions for the sham controls, vehicle-treated CCI rats and G-CSF-treated CCI rats were identical.

### 2.4. Western Blotting

We performed micro-Western array [41] at the National Health Research Institutes in Taiwan to screen for the expression of different proteins (Protein kinase B (Akt1), B-cell lymphoma 2 (Bcl-2), phospho-c-Jun, phospho-p44/p42, NF-κB, Bcl2-associated X protein [BAX], and cytochrome c) in the injured sciatic nerve. We then performed Western blotting to verify the protein expression and examined the expression of microtubule-associated protein light chain 3-II (LC3II) and cleaved caspase-3 in the injured sciatic nerve in our laboratory. Rats were anesthetized using pentobarbital (50 mg/kg body weight) and sacrificed. We quickly separated and collected the sciatic nerve at the CCI site (the site of the four chromic gut ligatures) in tissue lysis buffer (Tissue Protein Extraction Reagent, Thermo Scientific, 78510) containing a protease inhibitor cocktail (Roche, 11697498001) and phosphatase inhibitor (Roche, 4693159001). After homogenization, the tissue lysates were centrifuged at 13,500 rpm for 45 min at 4 °C. The purified proteins were then sent to the National Health Research Institutes (NHRI) for micro-Western array analysis (Akt1, Bcl-2, phospho-c-Jun, phospho-p44/p42, NF-κB, BAX, and cytochrome c) according to their instructions [41]. Statistical analyses of those proteins (Akt1, Bcl-2, phospho-c-Jun, phospho-p44/p42, NF-κB, BAX, and cytochrome c) were performed using the data from micro-Western array in the NHRI; then, we also verified the protein expression findings by traditional Western blotting in our laboratory. Analyzes of LC3II and cleaved caspase-3 expressions in the injured sciatic nerve were performed by traditional Western blotting in our laboratory. In our laboratory, protein samples were separated using sodium dodecyl sulfate-polyacrylamide gel electrophoresis (SDS-PAGE) and transferred onto polyvinylidene fluoride (PVDF) membranes. The blots were blocked overnight at 4 °C using bovine serum albumin (5%) (Bioshop, ALB001) in Tris-buffered saline (TBS) (Bioman, TBS101000) with 0.1% Tween-20 (Bionovas, AT1260-0500). The membranes were then incubated overnight at 4 °C with antibodies against LC3 (1:1000, MBL, PM036), BAX (1:500; Cell Signaling Technology, #2772), Bcl-2 (1:500; BD Transduction Laboratories, 610539), cytochrome c (1:500; GeneTex, GTX108585), cleaved caspase-3 (1:500; Cell Signaling Technology, #9662), Akt1/PKBα (1:500; Millipore, 05-796), phospho-p42/44 (1:500; Cell Signaling Technology, #4370), NF-κB (1:1000; Cell Signaling Technology, #6956), GAPDH (internal control; 1:6000; Proteintech, 60004-1) and β-actin (internal control; 1:6000; Cell Signaling Technology, #3700) in TBS containing 0.1% Tween-20 (Bionovas, AT1260-0500). This step was followed by incubation for 60 min at room temperature with horseradish peroxide (HRP)-linked secondary antibodies (1:10,000, anti-rabbit IgG; Cell Signaling, #7074. 1:10,000, anti-mouse IgG; Cell Signaling, #7076). All washing was performed with TBS containing 0.1% Tween-20. The bands were detected using micro-Western arrays and UVP Chemstudio software (Analytikjena). Each band was normalized to an internal control (actin or GAPDH). We compared the protein expression between the sham controls and experimental rats with or without G-CSF treatment. In the LC3 Western blot analyses, only the LC3II bands were used for quantification, in accordance with previously published studies [21,25,42,43]. According to the manufacturer’s instructions (anti-caspase-3 antibody (Cell Signaling Technology, #9662) and anti-phospho-p44/42 MAPK antibody (Cell Signaling Technology, #4370)), the two bands (p12 and p17 fragments) that are visible in the cleaved caspase-3 Western blot are both active forms of caspase-3, and the two bands that are visible in the phospho-p44/p42 Western blot indicate phospho-p44 and phospho-p42. As previously described, the bands in the cleaved caspase-3 and phospho-p44/p42 Western blots must be considered together for quantitative analyses [44,45].

### 2.5. Immunohistochemistry

The rats were deeply anesthetized using sodium pentobarbital and transcardially perfused with 4% paraformaldehyde. The injured sciatic nerves (the site of the four chromic gut ligatures) and the right L5/L6 DRG were resected and placed in 4% paraformaldehyde for 4 hours and incubated overnight in 30% sucrose at 4 °C. The samples were subsequently embedded in optimal cutting temperature (OCT) compound (Sakura, Tissue-Tek 4583) and rapidly frozen. We selected every fourth section in a series of 10 μm sections of the sciatic nerve and DRG for immunostaining. Tissue sections of the injured sciatic nerve and DRG were obtained using a freezing microtome (Leica, CM 3050) and then mounted on polylysine-coated slides. For double immunofluorescence analysis, all of the tissue sections were blocked with 5% normal goat serum (Invitrogen, 31872) or 5% normal donkey serum (Abcam, ab7475). The sections were then incubated overnight at 4 °C with antibodies against LC3 (1:500, MBL, PM036) and cleaved caspase-3 (1:500, Cell Signaling Technology, #9664) combined with antibodies against CGRP (1:1000, Abcam, ab36001), NF200 (1:500, Sigma-Aldrich, N0142), IB4 (Alexa Fluor 488 conjugates, Invitrogen, I-21411) and S100 (S100 β-Subunit; 1: 500; Sigma-Aldrich, AMAB91038), followed by incubation with fluorescent-conjugated secondary antibodies (1:1000; Jackson ImmunoResearch Laboratories, Alexa Fluor 488-conjugated goat anti-mouse, 115-545-003/donkey anti-goat, 705-545-003, and Alexa Fluor 594-conjugated goat anti-rabbit, 111-585-144/donkey anti- rabbit, 711-585-152). The images were acquired using a fluorescence microscope (Olympus, BX51) connected to a digital camera and computer; montages were created and analyzed using MetaMorph (version 7.8; Molecular Devices). Positively stained axons (LC3 + CGRP, LC3 + NF200, LC3 + S100) and positively stained DRG neurons (LC3 + NF200, LC3 + CGRP, LC3 + IB4, cleaved caspase-3 + NF200, cleaved caspase-3 + CGRP, cleaved caspase-3 + IB4) were preliminarily analyzed by MetaMorph software and confirmed manually by an investigator who was blinded to the status of the rats. We compared the number of positively stained axons and DRG neurons between the sham controls and experimental rats with or without G-CSF treatment.

### 2.6. Statistical Analyses

Statistical analyses were performed using Prism software (version 9; GraphPad Software, San Diego, CA, USA). All quantitative data are presented as the mean ± standard error of the mean (SEM). For behavioral experiments, the Shapiro–Wilk test was used to determine if the behavioral experimental data were normally distributed. Normally distributed data were analyzed by two-way repeated measures ANOVA, followed by a post hoc Tukey’s test to compare the difference between each group. For Western blot data and positive cell counts, the Shapiro–Wilk test was used to determine whether the ELISA and immunohistochemical data were normally distributed. Normally distributed data were analyzed by one-way analysis of variance (ANOVA) followed by a post hoc uncorrected Fisher's LSD to compare the difference between each group. If the data were not normally distributed, we used the Kruskal–Wallis, and Mann–Whitney rank-sum post hoc tests (two-tailed). The Bartlett test was used to determine if the groups had equal variances before performing ANOVA. If the data lacked equal variance, we used Brown–Forsythe and Welch’s post hoc tests. *P* values less than 0.05 were considered statistically significant.

## 3. Results

### 3.1. Early Systemic G-CSF Treatment Alleviated Mechanical Allodynia in CCI Rats from the 1st to the 7th Day after Nerve Injury

Mechanical allodynia was measured using the von Frey filament test (Figure 1; from the 1st to the 7th day), and there were significant decreases in the vehicle-treated CCI rats compared to the sham control rats from the 1st to the 7th day after nerve injury (two-way repeated measures ANOVA followed by a post hoc Tukey’s test, *p* < 0.01). Moreover, mechanical allodynia in G-CSF-treated CCI rats was significantly attenuated compared to vehicle-treated CCI rats from the 1st to the 7th day after nerve injury (*p* < 0.01).

### 3.2. G-CSF Downregulated Akt1, Bcl-2, and Phospho-c-Jun Protein Expression but Did Not Alter Phospho-p44/42 or Nuclear Factor Kappa B (NF-κB) Protein Expression in the Injured Sciatic Nerve on the 1st Day after Nerve Injury

Western blot analysis revealed significantly lower Akt1 protein levels (an upstream regulator of autophagy) in the injured sciatic nerve in G-CSF-treated CCI rats than in vehicle-treated rats (one-way ANOVA, post hoc Fisher's LSD test or Kruskal–Wallis test, post hoc Mann–Whitney rank-sum test, if appropriate, *p* < 0.05) on the 1st day after nerve injury (Figure 2A). Additionally, Western blot analysis revealed that vehicle-treated CCI rats had significantly higher Bcl-2 levels in the injured sciatic nerve than sham control rats (one-way ANOVA, post hoc Fisher's LSD test or Kruskal–Wallis test, post hoc Mann–Whitney rank-sum test, if appropriate, *p* < 0.05) on the 1st day after nerve injury. In contrast, G-CSF-treated CCI rats had significantly lower Bcl-2 levels in the injured sciatic nerve than vehicle-treated CCI rats on the 1st day after nerve injury (*p* < 0.05) (Figure 2B). There were significantly lower phospho-c-Jun protein levels in the injured sciatic nerve in G-CSF-treated CCI rats than in vehicle-treated rats (one-way ANOVA, post hoc Fisher's LSD test or Kruskal–Wallis test, post hoc Mann–Whitney rank-sum test, if appropriate, *p* < 0.05) on the 1st day after nerve injury. However, there was no significant difference in phospho-c-Jun protein levels in vehicle-treated CCI rats and sham controls on the 1st day after nerve injury (Figure 2C). Both vehicle- and G-CSF-treated CCI rats showed significantly higher phospho-p44/42 protein levels in the injured nerve than those of the sham controls (one-way ANOVA, post hoc Fisher’s LSD test or Kruskal–Wallis test, post hoc Mann–Whitney rank-sum test, if appropriate, *p* < 0.05). G-CSF-treated CCI rats had lower phospho-p44/42 protein levels in the injured nerve than the vehicle-treated rats, but the difference was not statistically significant (Figure 2D). Finally, there was no significant difference in NF-κB levels in the injured nerves in sham, vehicle-treated, and G-CSF-treated CCI rats at the different time points (Figure 2E).

### 3.3. G-CSF Treatment Upregulated Autophagic Activity in the Injured Sciatic Nerve and DRG Neurons at the Early Phase (1st Day) after Nerve Injury

Western blot analysis showed that LC3II protein (a key protein in autophagy pathway) levels in the injured sciatic nerve were significantly higher in G-CSF-treated CCI rats than in vehicle-treated CCI rats and sham controls on the 1st day after nerve injury (one-way ANOVA, post hoc Fisher's LSD test or Kruskal–Wallis test, post hoc Mann–Whitney rank-sum test, if appropriate, *p* < 0.05). However, there were no significant differences in LC3II protein levels in vehicle-treated CCI rats and sham controls on the 1st day after nerve injury. LC3II protein levels in the injured sciatic nerve were significantly higher in both vehicle- and G-CSF-treated CCI rats than in sham controls from the 3rd to the 7th day after nerve injury. However, there were no significant differences in LC3II protein levels in vehicle-treated CCI rats and G-CSF-treated CCI rats from the 3rd to the 7th day after nerve injury (Figure 3A).

Immunohistochemical analysis revealed that many LC3-positive fibers in the injured nerves in vehicle-treated and G-CSF-treated CCI rats were costained for S100 (a β-subunit of Schwann cells), Neurofilament 200 (NF200) (a marker of large-diameter myelinated nerve fibers), and calcitonin gene-related peptide (CGRP) (a marker of small- to medium-diameter myelinated and unmyelinated peptidergic nerve fibers) (Figure 3B–D). Moreover, there were more LC3 + CGRP-positive fibers in the injured sciatic nerves in G-CSF-treated rats than in vehicle-treated rats and sham controls on the 1st day after nerve injury (one-way ANOVA, post hoc Fisher's LSD test or Kruskal–Wallis test, post hoc Mann–Whitney rank-sum test, if appropriate, *p* < 0.05) (Figure 3F). However, there were no significant differences in the numbers of LC3 + S100-positive and LC3 + NF200-positive fibers in vehicle-treated CCI rats and G-CSF-treated rats (Figure 3E,G). G-CSF-treated CCI rats also exhibited a higher number of LC3 + CGRP- and LC3 + NF200-positive DRG neurons than vehicle-treated CCI rats and sham controls (one-way ANOVA, post hoc Fisher's LSD test or Kruskal–Wallis test, post hoc Mann–Whitney rank-sum test, if appropriate, *p* < 0.05) (Figure 4A,B,D,E). Interestingly, there were no significant differences in the numbers of LC3 + Isolectin IB4 (IB4)-positive (a marker of nonpeptidergic unmyelinated C fibers) DRG neurons in G-CSF-treated and vehicle-treated CCI rats (Figure 4C,F). Based on previous results, we concluded that G-CSF treatment mainly upregulated autophagy in neurons with small- to medium-diameter myelinated and unmyelinated peptidergic nerve axons (CGRP-positive cells) and neurons with large-diameter myelinated axons (NF200-positive cells).

### 3.4. G-CSF Treatment Suppressed Apoptotic Activity in the Injured Sciatic Nerve and DRG Neurons in the Late Phase (7 Days) after Nerve Injury

Western blot analysis showed significantly higher expression of different apoptotic proteins (BAX and cleaved caspase-3) in the injured sciatic nerve in vehicle-treated CCI rats than in sham controls on the 3rd and 7th days after nerve injury (one-way ANOVA, post hoc Fisher's LSD test, or Kruskal–Wallis test, post hoc Mann–Whitney rank-sum test, if appropriate, *p* < 0.01). However, G-CSF treatment significantly suppressed the increase in apoptotic protein levels on the 7th day after nerve injury (one-way ANOVA, post hoc Fisher's LSD test or Kruskal–Wallis test, post hoc Mann–Whitney rank-sum test, if appropriate, *p* < 0.05) (Figure 5A and Figure 6A). G-CSF treatment also downregulated cytochrome c in the injured sciatic nerve on the 7th day after nerve injury, but there was no statistically significant difference compared with that in the other groups (Figure 5B).

We stained cleaved caspase-3-positive neurons in the injured DRG for NF200, CGRP, and IB4 (Figure 6B–D). We found significantly more neurons that were positive for cleaved caspase-3 + CGRP in the injured DRG in vehicle-treated CCI rats than in sham controls (one-way ANOVA, post hoc LSD Fisher's test or Kruskal–Wallis test, post hoc Mann–Whitney rank-sum test, if appropriate, *p* < 0.05) on the 7th day after nerve injury (Figure 6F). In contrast, G-CSF-treated CCI rats had fewer neurons that were positive for cleaved caspase-3 + NF200 and cleaved caspase-3 + CGRP than vehicle-treated CCI rats (one-way ANOVA, post hoc Fisher's LSD test or Kruskal–Wallis test, post hoc Mann–Whitney rank-sum test, if appropriate, *p* < 0.01) on the 7th day after nerve injury (Figure 6E,F). However, there were no significant differences in the numbers of neurons that were positive for cleaved caspase-3 + IB4 in sham, vehicle-treated, and G-CSF-treated CCI rats (Figure 6G). These results showed that neurons with small- to medium-diameter myelinated and unmyelinated peptidergic nerve axons (CGRP-positive cells) and neurons with large-diameter myelinated axons neurons (NF200-positive cells), which exhibited upregulated autophagy in the early phase, exhibited decreased apoptosis in the late phase after nerve injury.

## 4. Discussion

Our results showed that autophagic activity in the injured sciatic nerve in vehicle-treated CCI rats was upregulated from the 3rd to the 7th day after nerve injury. Furthermore, G-CSF treatment enhanced the upregulated autophagic activity in the injured nerve on the 1st day after nerve injury, which correlated with neuropathic pain attenuation beginning on the 1st day after nerve injury (more significant beginning on the 3rd day after nerve injury). There was an increase in autophagic activity in the neurons with small- to medium-diameter myelinated peptidergic axons (CGRP-positive cells) and neurons with large-diameter myelinated axons (NF200-positive cells). These findings are consistent with those of previous studies, suggesting that autophagy in injured nerves and axons is a spontaneous protective consequence of nerve injury. Guo et al. showed increased LC3II activity in the ipsilateral DRG from the 3rd to the 21st day after spinal nerve ligation (SNL) injury [25]. Previous studies have also demonstrated that upregulated autophagic activity in injured nerves that is mediated by different treatments (rapamycin, metformin, HBO therapy, and even microRNA therapy) can attenuate neuropathic pain, while the downregulation of autophagic activity can increase neuropathic pain [23,24,25,28,29]. For example, Marinelli et al. injected a single dose of rapamycin (which upregulates autophagic activity) intraplantarly into the feet of CCI rats on the 3rd day after nerve injury, and mechanical allodynia was alleviated from the 4th to 14th day and the 28th to 35th day after nerve injury in CCI rats [24]. Guo et al. injected rapamycin into the L5 DRG before and on the 7th day after SNL, and the researchers reported that neuropathic pain was alleviated dose-dependently by rapamycin from the 1st day to the 4th day and the 7th to 12th day after nerve injury [25]. Ma et al. reported that intrathecal administration of modified citrus pectin, which suppresses neuroinflammation, attenuated mechanical allodynia by downregulating autophagic activity in SNL rats from the 2nd to 14th day after nerve injury [43]. Menzie-Suderam et al. reported that single G-CSF gene therapy can downregulate LC3II expression in different brain areas of mice that received bilateral common carotid artery occlusion (CCAO) on the 7th day after ischemic injury [14]. However, in those studies, the researchers found that this downregulation of autophagic activity occurred in the late phase (7 days and 10 days after injury) rather than the earlier phase after nerve injury. In fact, Ma et al. reported no significant changes in early phase autophagic activity after modified citrus pectin treatment [43]. Based on our results and those of the aforementioned studies [23,24,25,28,29], we suggest that the attenuation of mechanical allodynia via the upregulation of autophagic activity in the injured nerve is a time-dependent process. Under untreated conditions, autophagic activity in the injured nerve gradually increased from the 1st day and reached a peak on the 3rd to the 7th days after nerve injury. However, the use of pharmacologic methods such as G-CSF treatment to enhance autophagic activity in neurons in the early phase (before the 1st day) after nerve injury may help to attenuate neuropathic pain.

The underlying mechanisms by which autophagy regulates neuropathic pain are complex interactions between many proinflammatory cytokines [46]. Our previous studies showed that G-CSF can abrogate the increased proinflammatory cytokine (IL-6 and TNF-α) and chemokine (MCP-1) levels in the DRG from the 2nd to 6th day and 7th day after nerve injury, respectively [1,5]. Previous studies have also shown that autophagy can suppress proinflammatory cytokine activity [23,46,47,48]. For example, autophagy can inhibit IL-1 family cytokines [48]. Bussi et al. showed that autophagy downregulates the expression of proinflammatory cytokines in cultured microglial cells [47]. Shi et al. found that a microRNA-195 inhibitor attenuated neuropathic pain by upregulating autophagy and downregulating the expression of proinflammatory cytokines in the spinal cords of SNL rats. Moreover, 3-MA (an autophagy inhibitor) has been shown to reverse the downregulation of proinflammatory cytokines after microRNA-195 inhibitor treatment. That study showed that microRNA-195 inhibitors could suppress the expression of proinflammatory cytokines by upregulating autophagic activity [23]. It is well known that proinflammatory cytokines play important roles in neuropathic pain, and inhibiting proinflammatory cytokines may help to attenuate neuropathic pain [49]. Based on previous studies and our results, we proposed that elevated autophagic activity in the early phase after nerve injury could downregulate proinflammatory cytokine expression, further attenuating neuropathic pain. Moreover, the time of autophagy activation is a determinant factor in neuropathic pain treatment. Once proinflammatory cytokines reach a threshold level and induce neuropathic pain, increased autophagic activity cannot suppress pain development. In contrast, an increase in autophagic activity in the early phase before proinflammatory cytokines reach the threshold level and induce neuropathic pain can effectively alleviate pain development. This is consistent with our findings that vehicle-treated CCI rats also exhibited increased autophagic activity on the 3rd day after nerve injury but showed increases in proinflammatory cytokine/chemokine (IL-6, TNF-α, and MCP-1) levels in the DRG from the 2nd to 7th days [1,5] after nerve injury, which occurred before autophagic activity increased.

There is also ex vivo evidence that G-CSF directly activates autophagic activity in neutrophils and hematopoietic stem cells (HSCs) from both mice and human donors [15]. Guo et al. showed that G-CSF promoted autophagic activity in the spinal cords of rats with spinal cord injury from the 1st day after nerve injury by inhibiting the NF-κB signaling pathway [9]. Gomez-Sanchez et al. reported that autophagy in Schwann cells is mediated by the mechanistic target of rapamycin (mTOR)-independent c-Jun pathway [42]. However, in our study, we found no differences in the expression levels of phospho-c-Jun and NF-κB in the injured nerve in CCI rats with and without G-CSF treatment during the early phase after nerve injury. Instead, we found that there were significantly lower Akt1 (an upstream regulator of autophagy) and Bcl-2 protein levels in the injured sciatic nerve in G-CSF-treated CCI rats than in vehicle-treated rats on the 1st day after nerve injury. Decreased Akt1 protein levels may downregulate mTOR to further upregulate autophagic activity [50]. Although studies by Wu et al. showed that G-CSF treatment could upregulate phospho-Akt expression in different brain areas of mice that received bilateral common carotid artery occlusion (CCAO) injury, the authors found that the upregulation of phospho-Akt protein expression occurred in the late phase (7 days after CCAO) rather than the early phase after ischemic injury [13,14]. On the other hand, we found that the elevated Bcl-2 expression in the injured nerves of CCI rats was suppressed by G-CSF treatment on the 1st day after nerve injury. These findings were consistent with the results of a study by Komine-Kobayashi showing that G-CSF could upregulate antiapoptotic Bcl-2 in the brains of mice after transient focal ischemic injury at 24 and 72 hours after reperfusion [12]. Previous ex vivo cellular culture studies have shown that morphine induces Beclin-1-dependent autophagy by increasing Beclin-1 and suppressing Bcl-2 expression [51,52]. Our previous studies showed that G-CSF treatment upregulated endogenous opioid-containing PMNs and opioid receptors in the injured nerve from the 12th and 72nd hour after nerve injury, respectively [1,2]. Thus, G-CSF may also upregulate autophagy in the early phase after nerve injury by downregulating Bcl-2 expression through the upregulation of endogenous opioids/opioid receptors.

In addition to autophagy, apoptosis is another important cellular mechanism in stressed cells. It was reported that the expression of cytochrome c, cleaved caspase-3, and different caspase genes was increased in the DRG neurons of rats with spinal nerve crush injury [31], rats with sciatic nerve transection injury [32], and diabetic rats [34]. Furthermore, drugs that inhibit caspase activity have been shown to alleviate mechanical allodynia in rats with spared nerve injury (SNI) [53] and small fiber neuropathy [54]. In addition to upregulated autophagic activity in the early phase after nerve injury, our current study also showed that there was significant downregulation of apoptotic activity in the injured nerves and DRG neurons of G-CSF-treated CCI rats in the late phase (7th day) after nerve injury. These findings are consistent with those of studies by Wu that showed that G-CSF treatment could downregulate proapoptotic proteins and upregulate antiapoptotic proteins in different brain areas of mice that underwent bilateral common carotid artery occlusion (CCAO) injury at 7 days after ischemic injury [13,14].

Jacobs and Ro reported significant damage to large and small myelinated fibers but not unmyelinated fibers in the CCI model [55]. In the present study, we found similar results that showed that the apoptotic activities in both large-diameter myelinated (NF200-positive) and small- to medium-diameter myelinated and unmyelinated peptidergic (CGRP-positive) neurons after nerve injury were suppressed by G-CSF treatment. However, there were no significant differences in the numbers of nonpeptidergic unmyelinated nociceptive C-fiber (IB4 positive) neurons between sham control, vehicle-treated, and G-CSF-treated CCI rats. Interestingly, after G-CSF treatment, autophagic activity increased mainly in large-diameter myelinated (NF200-positive) and small- to medium-diameter myelinated and unmyelinated peptidergic (CGRP-positive) DRG neurons but not in nonpeptidergic unmyelinated nociceptive C-fiber (IB4-positive) DRG neurons in the early phase after nerve injury. Based on these results and those of previous studies, we concluded that neuropathic pain could be attenuated by upregulating autophagic activity and inhibiting apoptotic activity in the injured nerves and DRG neurons of G-CSF-treated CCI rats in the early and late phases after nerve injury, respectively; moreover, these effects mainly occurred in NF200- and CGRP-positive neurons.

Autophagy and apoptosis exhibit some degree of mutual inhibition [56]. Autophagy and apoptosis continuously communicate with each other through several proteins, including p62, Bcl-2, Beclin-1, and caspase-3 [57]. For example, the apoptotic protein caspase-3 inhibits autophagy by cleaving Beclin-1 [57]. On the other hand, Bcl-2 inhibits both autophagy and apoptosis [57]. In this study, we did not find significant temporal correlations between these proteins and autophagy and apoptosis. Instead, we concluded that elevated autophagy in the early phase after nerve injury can downregulate proinflammatory cytokine expression in the late phase after nerve injury. Furthermore, proinflammatory cytokines, including IL-1 and TNF-α, were reported to mediate apoptosis by increasing glutamate levels in neural cultures [58]; in contrast, a TNF-α-neutralizing antibody was reported to inhibit apoptosis in cultured DRG neurons [59]. Based on the observed sequence of events, we hypothesize that G-CSF treatment initially upregulates autophagic activity, then inhibits the expression of proinflammatory cytokines, and thereafter downregulates apoptotic activity. The findings of this study suggest that early activation of autophagy by G-CSF before the increase in proinflammatory cytokine levels plays a determinant role in neuropathic pain. Moreover, these findings also suggest that autophagy-mediated attenuation of neuropathic pain is a time-dependent process. In addition to the previously described mechanisms, G-CSF has been shown to exert direct antiapoptotic effects on cardiomyocytes [60,61]. In previous ex vivo studies, G-CSF directly suppressed apoptotic activity in neutrophils by delaying extracellular calcium influx, which activates caspase-3 [16]. G-CSF can also modulate different microRNA expressions [5,62]. Furthermore, microRNAs play an important role in autophagy regulations [63], and even in the crosstalk between autophagy and apoptosis [64]. For example, Li’s study has showed that microRNA-378 promotes autophagy but inhibits apoptosis in skeletal muscle [65]. Our previous study had shown that G-CSF can upregulate the decreased microRNA-122 expressions in the dorsal root ganglia at the early phase after nerve injury, then the upregulated microRNA-122 can suppress monocyte chemoattractant protein-1 (MCP-1) expressions, which further attenuate neuropathic pain [5]. Wang’s study has shown that microRNA-122 can promote autophagic activities in the hepatocytes under arsenic stress [66]. Thus, G-CSF treatment probably also upregulated autophagic activities through upregulating microRNA-122 levels.

## 5. Conclusions

Based on our results and those of previous studies, we concluded that G-CSF is a multipotent agent for treating neuropathic pain that functions through different signaling pathways, including the upregulation of MOR expression and autophagic activity through the suppression of Bcl-2 expression in the injured nerve in the early phase. This effect is followed by the suppression of proinflammatory cytokine expression in DRG neurons and the spinal dorsal horn and the subsequent downregulation of apoptosis in DRG neurons, which suppresses microglial activation in the spinal dorsal horn in the late phase after nerve injury, thus attenuating neuropathic pain (Figure 7). Our findings suggest that medications that can modulate proinflammatory cytokine expression, autophagy, and apoptosis, such as G-CSF, have the potential to treat neuropathic pain. However, the timing of administration, dose, and indications for G-CSF treatment in cases of neuropathic pain still need to be validated by a large series of clinical studies.

## Figures and Tables

**Figure 1 biomedicines-09-00542-f001:**
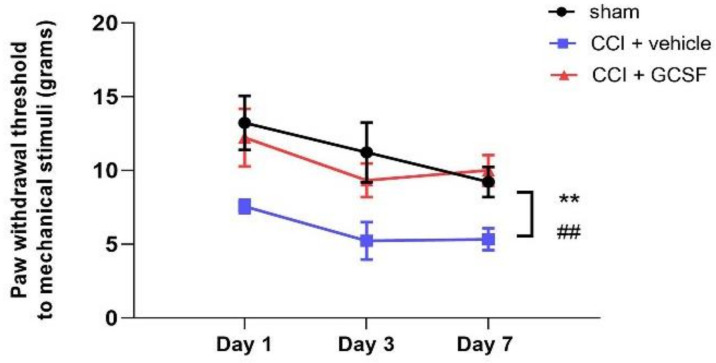
Early systemic G-CSF treatment alleviated mechanical allodynia in rats with chronic constriction injury (CCI) from the 1st to the 7th days after nerve injury. Significant mechanical allodynia developed in vehicle-treated CCI rats compared to sham control rats from the 1st to the 7th day after nerve injury (^##^
*p* < 0.01: vehicle-treated CCI rats compared to sham-operated control rats). In contrast, early G-CSF treatment alleviated mechanical allodynia from the 1st to the 7th day after nerve injury compared to that in vehicle-treated CCI rats (** *p* < 0.01: G-CSF-treated CCI rats compared to vehicle-treated CCI rats). The data are shown as the mean ± SEM. *n* = 9 in each group.

**Figure 2 biomedicines-09-00542-f002:**
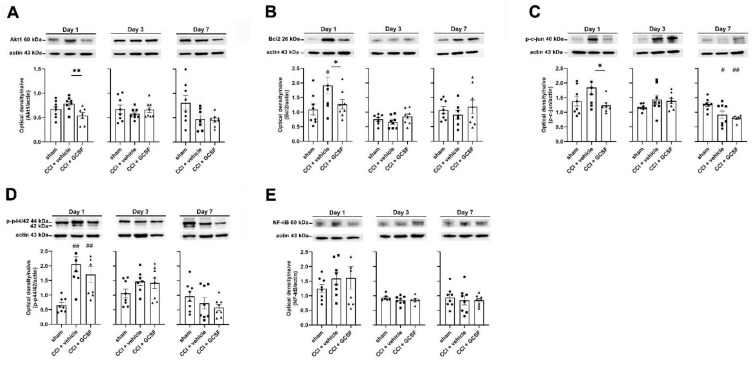
G-CSF downregulated Akt1, Bcl-2, and phospho-c-Jun protein levels in the injured sciatic nerve on the 1st day after nerve injury. However, G-CSF did not alter NF-κB and phospho-p44/42 protein expression in the injured sciatic nerve from the 1st day to the 7th day after nerve injury. Statistical analyses of those proteins (Akt1, Bcl-2, phospho-c-Jun, phospho-p44/p42, NF-κB) were analyzed by the data from micro-Western array in the NHRI, then we verified the protein expressions by the traditional representative Western blot bands in our laboratory. (**A**) Western blot analysis revealed significantly lower Akt1 protein levels (an upstream regulator of autophagy) in the injured sciatic nerve in G-CSF-treated CCI rats than in vehicle-treated CCI rats (** *p* < 0.01: G-CSF-treated CCI rats compared to vehicle-treated CCI rats) on the 1st day after nerve injury. However, there was no significant difference between vehicle-treated CCI rats and sham control rats. The data are shown as the means ± SEM. *n* = 8 in each group. (**B**) Western blot analysis revealed significantly higher Bcl-2 levels in the injured sciatic nerve in vehicle-treated CCI rats than in sham control rats (^#^
*p* < 0.05: vehicle-treated group compared to sham control rats) on the 1st day after nerve injury. In contrast, significantly lower Bcl-2 levels were observed in the injured sciatic nerve in G-CSF-treated CCI rats than in vehicle-treated CCI rats on the 1st day after nerve injury (* *p* < 0.05: G-CSF-treated CCI rats compared to vehicle-treated CCI rats). The data are shown as the means ± SEM. *n* = 8 in each group. (**C**) Western blot analysis revealed significantly lower phospho-c-Jun protein levels (an upstream regulator of autophagy) in the injured sciatic nerve in G-CSF-treated CCI rats than in vehicle-treated CCI rats (* *p* < 0.05: G-CSF-treated CCI rats compared to vehicle-treated CCI rats) on the 1st day after nerve injury. However, there was no significant difference between vehicle-treated CCI rats and sham control rats on the 1st day after nerve injury. Vehicle- and G-CSF-treated CCI rats had significantly lower phospho-c-Jun protein levels in the injured nerve than sham control rats (_#_
*p* < 0.05, _##_
*p* < 0.01: vehicle-treated CCI rats and G-CSF-treated CCI rats compared to sham controls) on the 7nd day after nerve injury. However, there was no significant difference between vehicle-treated rats and G-CFS-treated CCI rats. The data are shown as the means ± SEM. *n* = 8 in each group. (**D**) Vehicle- and G-CSF-treated CCI rats had significantly higher phospho-p44/42 protein levels in the injured nerve than sham control rats (^#^
*p* < 0.05, ^##^
*p* < 0.01: vehicle-treated CCI rats and G-CSF-treated CCI rats compared to sham controls) on the 1st day after nerve injury. Phospho-p44/42 protein levels were lower in the injured nerve in G-CSF-treated CCI rats than in vehicle-treated CCI rats on the 7th day after nerve injury; however, the difference was not statistically significant. The data are shown as the means ± SEM. *n* = 8 in each group. (**E**) Western blot analysis did not reveal significant changes in NF-κB protein expression in the injured nerve in sham control rats, vehicle-treated CCI rats, and G-CSF-treated CCI rats at different time points. The data are shown as the means ± SEM. *n* = 8 in each group.

**Figure 3 biomedicines-09-00542-f003:**
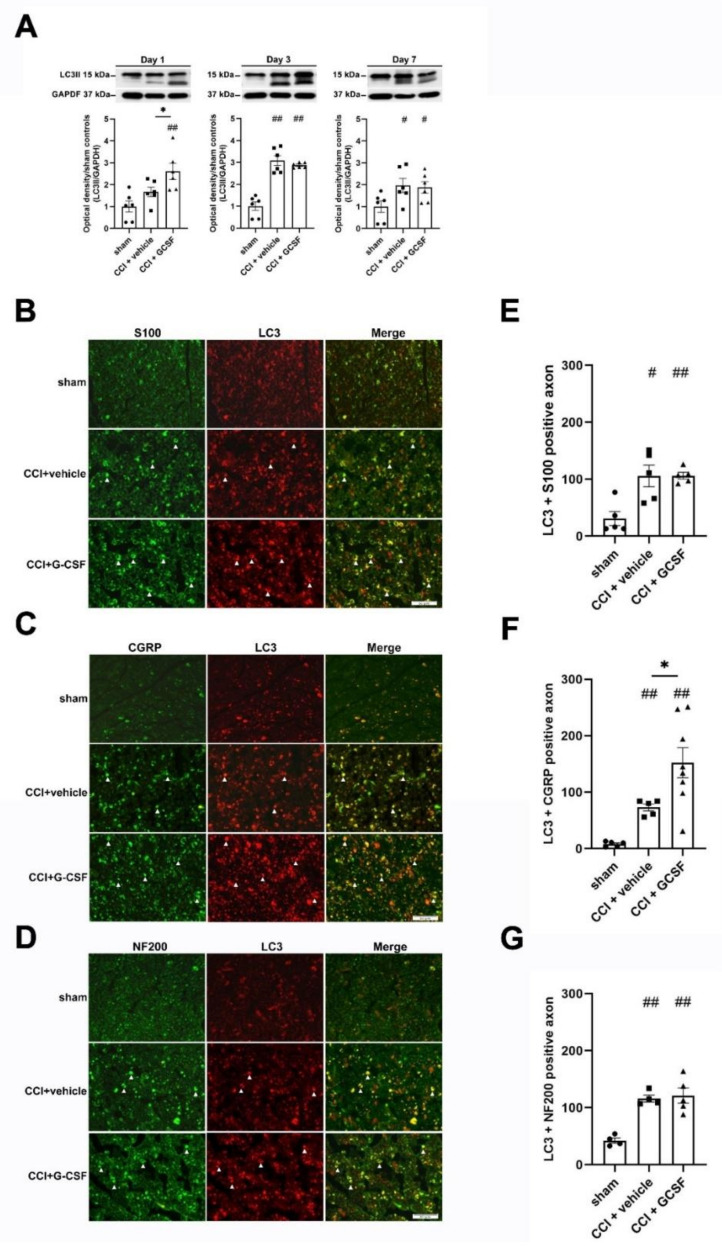
G-CSF upregulated autophagy in the injured sciatic nerve on the 1st day after nerve injury. (**A**) Western blot analysis revealed significantly higher LC3II protein levels in the injured sciatic nerve in G-CSF-treated CCI rats than in vehicle-treated CCI rats and sham control rats on the 1st day after nerve injury (* *p* < 0.05, ^##^
*p* < 0.01: G-CSF-treated CCI rats compared to vehicle-treated CCI rats and sham control rats). However, there was no significant difference in LC3II protein levels in vehicle-treated CCI rats and sham control rats on the 1st day after nerve injury. LC3II protein levels in the injured sciatic nerve were significantly higher in vehicle- and G-CSF-treated CCI rats than in sham control rats from the 3rd to the 7th day after nerve injury. (^##^
*p* < 0.01, ^#^
*p* < 0.05: the G-CSF-treated and vehicle-treated groups compared to the sham controls on the 3rd and 7th days after nerve injury). However, there was no significant difference between vehicle-treated CCI rats and GCSF-treated CCI rats. The data are shown as the means ± SEM. *n* = 6 in each group. (**B**–**D**) Representative images showing LC3-positive fibers (white arrowhead) in the injured sciatic nerve in sham control rats, vehicle-treated CCI rats and G-CSF-treated CCI rats on the 1st day after nerve injury. LC3-positive fibers were costained with S100 (a marker of the β-subunit of Schwann cells), NF200 (a marker of large-diameter myelinated axons) and CGRP (a marker of small- to medium-diameter myelinated and unmyelinated peptidergic axons). Scale bars = 50 µm. (**E**–**G**) Significantly higher numbers of axons that were positive for LC3 + S100 were observed in G-CSF-treated CCI rats and vehicle-treated CCI rats than in sham control rats on the 1st day after nerve injury. (^##^
*p* < 0.01, ^#^
*p* < 0.05: the G-CSF-treated CCI and vehicle-treated CCI groups compared to the sham controls). However, there was no significant difference between GCSF-treated CCI rats and vehicle-treated CCI rats. Significantly higher numbers of axons that were positive for LC3 + NF200 were also observed in G-CSF-treated CCI rats and vehicle-treated CCI rats than in sham control rats on the 1st day after nerve injury. (^##^
*p* < 0.01, ^##^
*p* < 0.01: the G-CSF-treated CCI and vehicle-treated CCI groups compared to the sham controls). However, there was no significant difference between GCSF-treated CCI rats and vehicle-treated CCI rats. G-CSF-treated CCI rats showed significantly higher numbers of LC3 + CGRP-positive axons than sham control rats and vehicle-treated CCI rats on the 1st day after nerve injury (* *p* < 0.05: the G-CSF-treated CCI group compared to the vehicle-treated CCI group; ^##^
*p* < 0.01: the G-CSF-treated CCI group compared to the sham controls). There were also significantly higher numbers of LC3 + CGRP-positive axons in the vehicle-treated CCI group than in the sham controls. (^##^
*p* < 0.01: the vehicle-treated CCI group compared to the sham controls). The data are shown as the means ± SEM. *n* = 5 in each group.

**Figure 4 biomedicines-09-00542-f004:**
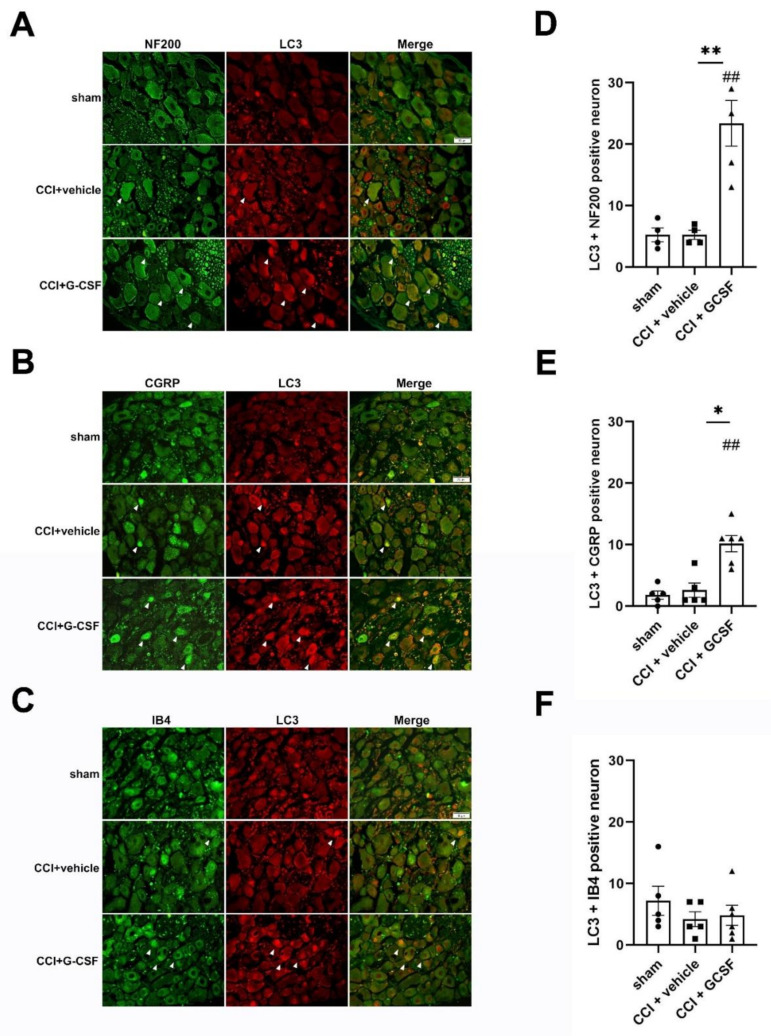
G-CSF upregulated LC3 protein levels in dorsal root ganglia neurons on the 1st day after nerve injury. (**A**–**C**) Representative images showing LC3 + NF200-, LC3 + CGRP-, and LC3 + IB4-positive DRG neurons (white arrowhead) in sham control rats, vehicle-treated CCI rats, and G-CSF-treated CCI rats on the 1st day after nerve injury. (**D**–**F**) Significantly higher numbers of neurons positive for LC3 + NF200 and LC3+ CGRP were observed in G-CSF-treated CCI rats than in vehicle-treated CCI rats and sham control rats (** *p* < 0.01, * *p* < 0.05: the G-CSF-treated CCI group compared to the vehicle-treated CCI group; ^##^
*p* < 0.01: the G-CSF-treated CCI group compared to the sham controls). However, there was no significant difference in the numbers of neurons that were positive for LC3 + NF200 and LC3 + CGRP between vehicle-treated CCI rats and sham control rats. There were also no significant differences in the numbers of LC3 + IB4-positive neurons between G-CSF-treated CCI rats, vehicle-treated CCI rats, and sham control rats. Scale bars = 50 µm. The data are shown as the means ± SEM. *n* = 5 in each group.

**Figure 5 biomedicines-09-00542-f005:**
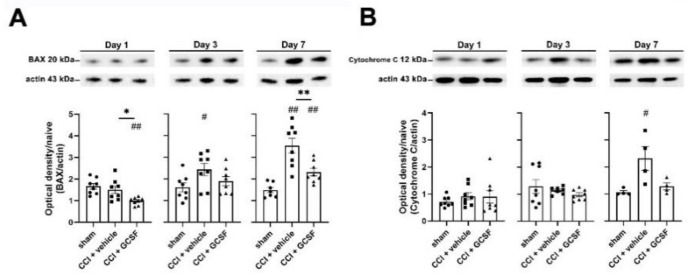
G-CSF downregulated BAX protein levels and cytochrome c (apoptosis) in the injured sciatic nerve on the 7th day after nerve injury. Statistical analyses of those proteins (BAX, cytochrome c) were analyzed by the data from micro-Western array in the NHRI; then, we verified the protein expressions by the traditional representative Western blot bands in our laboratory. (**A**) Western blot analysis revealed significantly lower BAX levels in the injured sciatic nerve in G-CSF-treated CCI rats than in vehicle-treated CCI rats and sham control rats (* *p* < 0.05: G-CSF-treated CCI rats compared to vehicle-treated CCI rats; ^##^
*p* < 0.01: G-CSF-treated CCI rats compared to sham control rats) on the 1st day after nerve injury. However, there was no significant difference between vehicle-treated CCI rats and sham control rats. In contrast, western blot analysis revealed significantly higher BAX levels in the injured sciatic nerve in vehicle-treated CCI rats than in sham control rats (^##^
*p* < 0.01, ^#^
*p* < 0.05: vehicle-treated and G-CSF-treated CCI rats compared to sham control rats) on the 3rd and 7th days after nerve injury. Significantly lower BAX levels were observed in the injured sciatic nerve in G-CSF-treated CCI rats than in vehicle-treated CCI rats on the 7th day after nerve injury (** *p* < 0.01: G-CSF-treated CCI rats compared to vehicle-treated CCI rats). The data are shown as the means ± SEM. *n* = 8 for each group. (**B**) On the 7th day after nerve injury, there were higher cytochrome c levels in the injured nerve in vehicle-treated CCI rats than in sham control rats (^#^
*p* < 0.05: vehicle-treated and G-CSF-treated CCI rats compared to sham control rats). However, there were no significant differences between the vehicle-treated group and the G-CSF-treated group. The data are shown as the means ± SEM. *n* = 8 for each group.

**Figure 6 biomedicines-09-00542-f006:**
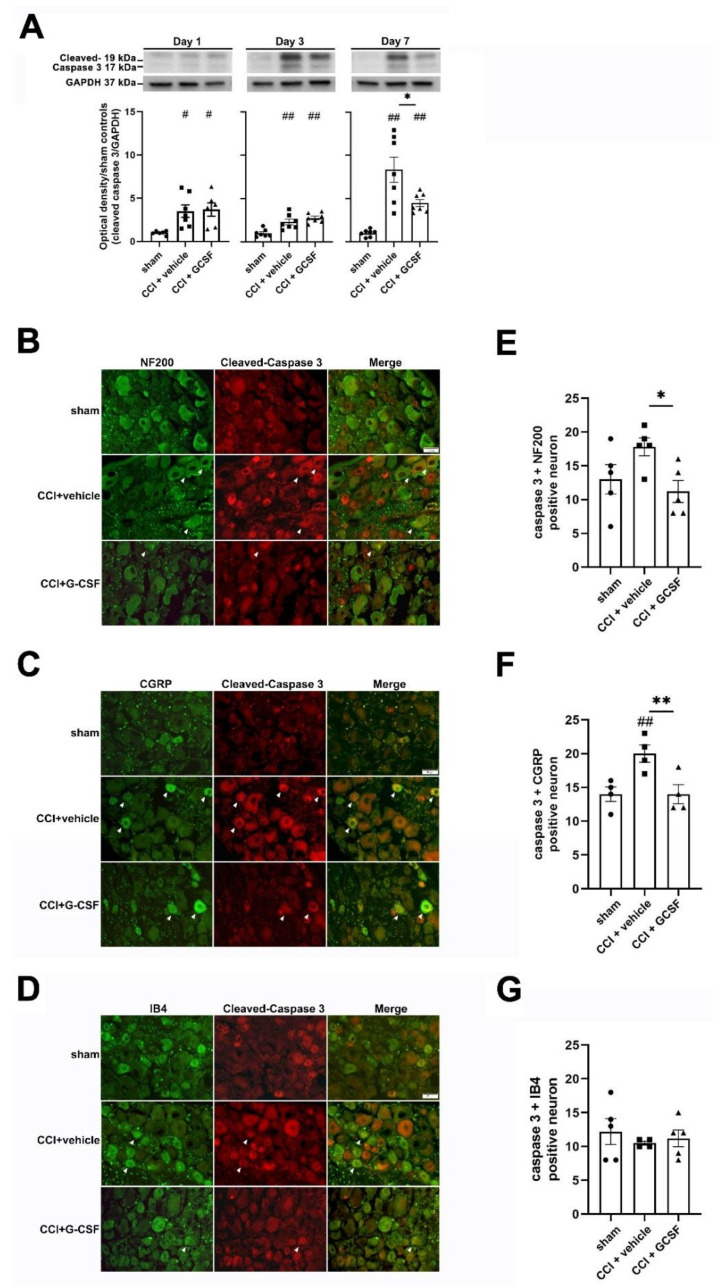
G-CSF downregulated cleaved caspase-3 (apoptotic) protein levels in dorsal root ganglia neurons on the 7th day after nerve injury. (**A**) Western blot analysis revealed significantly higher cleaved caspase-3 levels in the injured sciatic nerve in vehicle-treated CCI rats than in sham control rats (^##^
*p* < 0.01, ^#^
*p* < 0.05: vehicle-treated and G-CSF-treated CCI rats compared to sham control rats) from the 1st to the 7th day after nerve injury. In contrast, significantly lower cleaved caspase-3 levels were observed in the injured sciatic nerve in G-CSF-treated CCI rats than in vehicle-treated CCI rats on the 7th day after nerve injury (* *p* < 0.05: G-CSF-treated CCI rats compared to vehicle-treated CCI rats). The data are shown as the means ± SEM. *n* = 6 for each group. (**B**–**D**) Representative images showing cleaved caspase-3 + NF200-, cleaved caspase-3 + CGRP-, and cleaved caspase-3 + IB4-positive DRG neurons (white arrowhead) in sham control rats, vehicle-treated CCI rats, and G-CSF-treated CCI rats on the 7th day after nerve injury. Scale bars = 50 µm. (**E**–**G**) Significantly higher numbers of neurons that were positive for cleaved caspase-3 and CGRP were observed in vehicle-treated CCI rats than in sham control rats (^##^
*p* < 0.01: the vehicle-treated group compared to the sham controls). Furthermore, significantly lower numbers of neurons that were positive for cleaved caspase-3 and NF200/CGRP were observed in G-CSF-treated CCI rats than in vehicle-treated CCI rats (** *p* < 0.01, * *p* < 0.05: the G-CSF-treated group compared to vehicle-treated group). There was no significant difference in the numbers of neurons that were positive for cleaved caspase-3 and IB4 in sham control rats, vehicle-treated CCI rats, and G-CSF-treated CCI rats. The data are shown as the means ± SEM. *n* = 5 for each group.

**Figure 7 biomedicines-09-00542-f007:**
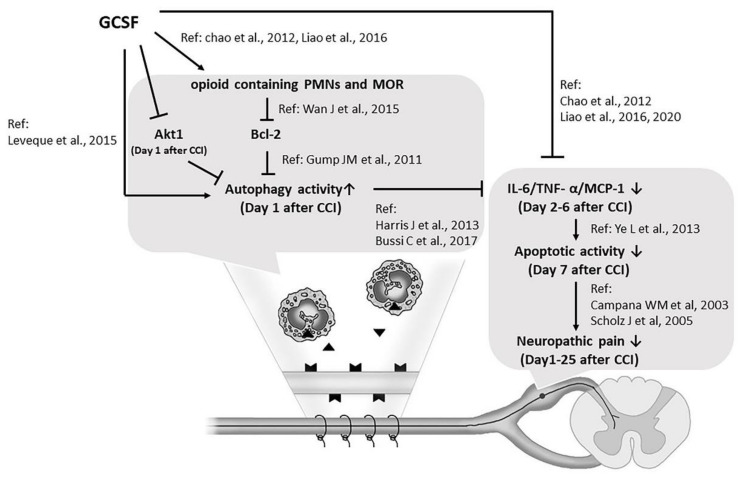
G-CSF is a multipotent agent for treating neuropathic pain. G-CSF upregulates MOR expression and autophagic activity in the injured nerve in the early phase, suppresses proinflammatory cytokine expression in DRG neurons, and thereafter downregulates apoptotic activity in DRG neurons, thus attenuating neuropathic pain. (▼, Opioid; 
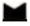
, Mu opioid receptor; ↑: increase; ↓: decrease; →: enhance; ―|: suppress).

## Data Availability

The data presented in this study are available in this manuscript.

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
