# Peer review of "Interactions between Autophagy, Proinflammatory Cytokines, and Apoptosis in Neuropathic Pain: Granulocyte Colony Stimulating Factor as a Multipotent Therapy in Rats with Chronic Constriction Injury"

_biomedicines, 2021, doi:10.3390/biomedicines9050542_

Round 1
Reviewer 1 Report
This manuscript reports how GCSF could potentially be used for reduced neuropathic in peripheral nerves via altering autophagic activity. This manuscript is of interest because it reveals the interplay between autophagy, apoptosis and cytokine mediation and demonstrates how modifying the various responses can modulate the neural responses. Treatments that can reduce neuropathic pain are obviously of interest to the field and while GCSF itself may be useful, it opens up the possibility that other potential drugs could also target this pathway and therefore offer clinical benefits.
Overall, the manuscript is well written and the results are presented well with good descriptions of the figures and analyses.
The Western blot analyses of Fig.2 are generally fine, however there are a few images of blots that do not correspond to the quantification:
Fig. 2C, Day 1, CCI+GCSF: the blot above has a very low actin band and relatively strong c-jun band but the quantification shows a shorter column than the vehicle treatment. Is this correct?
Fig. 2C, Day 3, CCI+GCSF: the strong actin band would indicate that the c-jun was relatively lower, especially compared to the vehicle treatment but the quantification shows otherwise.
Fig. 2D, Day 3, vehicle treatment: the strong actin band would indicate that the AKT1 would be relatively lower, but the quantification shows otherwise.
Fig. 2E, Day 1, CCI+GCSF: the Bcl2:actin band ratio appears to show stronger actin, particularly when compared to the vehicle but the quantification shows similar levels.
Can you authors provide the blots for all replicates and confirm the quantifications.
For Figs. 3 and 4 the resolution of the image are not sufficient to determine the quality of the immunohistochemistry. Can higher resolution images be provided.
In the Discussion, line 459, “on the 3rd to 5th days…” should be “3rd to 7th days…”
There a couple of minor English expression errors in the abstract and Introduction which careful proofreading will pick up.
Author Response
Thanks for your careful checking and constructive comments. The western blot analyses of autophagic protein (LC3) and apoptotic protein (cleaved caspase 3) were performed at our laboratory after several repeated experiments that showed similar results, and thus we are quite confident that G-CSF can upregulate autophagic activities at the early phase and downregulate apoptotic activities at the late phase after nerve injury. The purified proteins were then sent to the National Health Research Institutes (NHRI) for micro-western array analysis (Akt1, Bcl-2, phospho-c-Jun, phospho-p44/p42, NF-κB, BAX, and cytochrome c) according to their instructions (Figure 2) to elucidate more detailed mechanisms. Statistical analyses of those proteins (Akt1, Bcl-2, phospho-c-Jun, phospho-p44/p42, NF-κB, BAX, and cytochrome c) were analyzed by the data from micro-western array at the NHRI, then we further verified the protein expressions and chose the representative western blot bands by traditional western blotting in our laboratory (page 3; lines 131-134). We tried to provide all original data in detail in the manuscript, and there are several studies that can confirm our western blot findings (Bcl-2 activities were downregulated by opioid treatment)[1]. In the revised manuscript, we have provided better representative western blot images of Figure 2. We have also provided original micro-western array pictures from NHRI in the attached files to reviewers to verify that whether the image qualities could fulfill the standard of publication.
We have also provided higher resolution images of Figures 3 and 4, and corrected line 459, “on the 3rd to 5th days…” to “3rd to 7th days…”. Some English expression errors in the abstract and introduction were also corrected (lines 22, 33, 45, 71).
Reference:
- Wan, J.; Ma, J.; Anand, V.; Ramakrishnan, S.; Roy, S. Morphine potentiates LPS-induced autophagy initiation but inhibits autophagosomal maturation through distinct TLR4-dependent and independent pathways. Acta Physiol (Oxf) 2015, 214, 189-199, doi:10.1111/apha.12506.

Reviewer 2 Report
- It is unclear why the baseline for CCI+GCSF turns out to be lower than sham and CCI+vehicle in figure 1. Some explanations are required to interpret the data.
- Authors should include acetone test as another behavioral modality since CCI is a neuropathic pain condition.
- It is unclear why the beta actin controls are not consistent in Figure 2 Western blots. Beta actin was used as a positive control and its expression should not be varying with the downregulation (Akt1, Bcl-2, and phospho-c-Jun) or unaltered protein levels of NF-κB and phospho-p44/42 p after G-CSF treatment. Please provide plausible explanation or repeat a few Western blots, such as Figure 2B and 2C.
- Please zoom in the inset images in figures 3 and 4. It is hard to see the upregulated protein markers.
Author Response
Regarding to your meticulous questions and constructive suggestions. Our responses are as explained below.
- In Figure 1, the paw withdrawal threshold to mechanical stimuli (grams) at day 1 after nerve injury of sham, CCI + vehicle, and CCI + GCSF groups were 13.22 +/- 1.832, 7.56 +/- 0.44, and 12.22 +/- 1.95, respectively. CCI + GCSF group had higher withdraw thresholds compared to CCI + vehicle group from the 1st to the 7th days after nerve injury (**P < 0.01: G-CSF-treated CCI rats compared to vehicle-treated CCI rats by two-way repeated measures ANOVA). However, the withdraw thresholds of the CCI + GCSF group are similar to those of the sham group from the 1st to the 7th days after nerve injury (P > 0.05: G-CSF-treated CCI rats compared to sham rats by two-way repeated measures ANOVA).
- We have done the thermal hyperalgesia tests that were performed by 46o C hot water bath in our previous publication [1]. The results have confirmed that GCSF can also alleviate thermal hyperalgesia. However, we didn’t perform acetone test in the current study.
- The western blot analyses of autophagic protein (LC3) and apoptotic protein (cleaved caspase 3) were performed in our laboratory after several repeated experiments that showed similar results, and thus we are quite confident that G-CSF can upregulate autophagic activities at the early phase and downregulate apoptotic activities at the late phase after nerve injury. The purified proteins were then sent to the National Health Research Institutes (NHRI) for micro-western array analysis (Akt1, Bcl-2, phospho-c-Jun, phospho-p44/p42, NF-κB, BAX, and cytochrome c) according to their instructions (Figure 2) to elucidate more detailed mechanisms. Statistical analyses of those proteins (Akt1, Bcl-2, phospho-c-Jun, phospho-p44/p42, NF-κB, BAX, and cytochrome c) were analyzed by the data from micro-western array at the NHRI, then we further verified the protein expressions and chose the representative western blot bands by traditional western blotting in our laboratory (page 3; lines 131-134). We tried to provide all original data in detail in the manuscript, and there are several studies that can confirm our western blot findings (Bcl-2 activities were downregulated by opioid treatment)[2]. In the revised manuscript, we have provided better representative western blot images of Figure 2. We have also provided original micro-western array pictures from NHRI in the attached files to reviewers to verify that whether the image qualities could fulfill the standard of publication.
- We have also provided higher resolution images of Figures 3 and 4 in the revised version.
Reference:
- Chao, P.K.; Lu, K.T.; Lee, Y.L.; Chen, J.C.; Wang, H.L.; Yang, Y.L.; Cheng, M.Y.; Liao, M.F.; Ro, L.S. Early systemic granulocyte-colony stimulating factor treatment attenuates neuropathic pain after peripheral nerve injury. PLoS One 2012, 7, e43680, doi:10.1371/journal.pone.0043680.
- Wan, J.; Ma, J.; Anand, V.; Ramakrishnan, S.; Roy, S. Morphine potentiates LPS-induced autophagy initiation but inhibits autophagosomal maturation through distinct TLR4-dependent and independent pathways. Acta Physiol (Oxf) 2015, 214, 189-199, doi:10.1111/apha.12506.

Reviewer 3 Report
The manuscript entitled "Interactions Between Autophagy, Proinflammatory Cytokines, and Apoptosis in Neuropathic Pain: Granulocyte Colony Stimulating Factor as a Multipotent Therapy in Rats with Chronic
Constriction Injury" is presented for peer review. The authors studied the behaviors of CCI rats with or without G-CSF treatment and the various levels of autophagigic, proinflammatory, and apoptotic markers in injured sciatic nerves and DRG neurons. Many clinical trials have demonstrated the analgesic effects of G-CSF in patients with compressive myelopathy.
The manuscript is well-structured and have good reference list. I have several concerns about this paper.
Major concerns: I propose to do flow cytometry analysis of primary cells derived from tissues to further evidence of G-CSF impact on apoptosis and autophagy in CCI model. Second, to further elucidate the possible mechanism, I recommend extracting RNA from nerves and perform qPCR to quantify gene expression (Bcl-2, Bcl-xl, caspase-3) after damage to dissect possible pathways. Also, transcriptional analysis of G-CCF regulation would be useful.
Third, I strongly recommend including speculation about miRNA regulation of autophagy in Discussion part.
Minor changes: colony-stimulating p.2 line 45
Please provide bars in Fig.3 and Fig.4 P.9-10 .
Author Response
Thanks for your constructive comments and suggestions. We will try to perform flow cytometry analysis of primary cells derived from tissues to provide further evidence of G-CSF impact on apoptosis and autophagy in CCI model in the future. In fact, we have found that opioid containing neutrophils increased in the tissue fluid around the injured nerves after GCSF treatment by flow cytometry analysis in our previous study [1]. We will also try to analyze the different autophagic and apoptotic gene expressions on injured nerve by qPCR methods in the future. We have added some discussion about the miRNA regulation of autophagy and apoptosis in the revised discussion part (page 17; lines 556-586). “G-CSF also can modulate different microRNA expressions[2,3]. Furthermore, microRNAs play an important role in autophagy regulations[4], and even in the crosstalk between autophagy and apoptosis[5]. For example, Li’s study has showed that microRNA-378 promotes autophagy but inhibits apoptosis in skeletal muscle[6]. Our previous study has shown that G-CSF can upregulate the decreased microRNA-122 expressions in the dorsal root ganglia at the early phase after nerve injury, then the upregulated microRNA-122 can suppress monocyte chemoattractant protein-1 (MCP-1) expressions, which further attenuate neuropathic pain[3]. Wang’s study has shown that microRNA-122 can promote autophagic activities in the hepatocytes under arsenic stress[7]. Thus, G-CSF treatment probably also upregulated autophagic activities through upregulating microRNA-122 levels.”
We have corrected “colony stimulating” to “colony-stimulating” in line 45. And we have provided bars in Figures 3 and 4.
Reference:
- Chao, P.K.; Lu, K.T.; Lee, Y.L.; Chen, J.C.; Wang, H.L.; Yang, Y.L.; Cheng, M.Y.; Liao, M.F.; Ro, L.S. Early systemic granulocyte-colony stimulating factor treatment attenuates neuropathic pain after peripheral nerve injury. PLoS One 2012, 7, e43680, doi:10.1371/journal.pone.0043680.
- Baez, A.; Martin-Antonio, B.; Piruat, J.I.; Prats, C.; Alvarez-Laderas, I.; Barbado, M.V.; Carmona, M.; Urbano-Ispizua, A.; Perez-Simon, J.A. Granulocyte colony-stimulating factor produces long-term changes in gene and microRNA expression profiles in CD34+ cells from healthy donors. Haematologica 2014, 99, 243-251, doi:10.3324/haematol.2013.086959.
- Liao, M.F.; Hsu, J.L.; Lu, K.T.; Chao, P.K.; Cheng, M.Y.; Hsu, H.C.; Lo, A.L.; Lee, Y.L.; Hung, Y.H.; Lyu, R.K., et al. Granulocyte Colony Stimulating Factor (GCSF) Can Attenuate Neuropathic Pain by Suppressing Monocyte Chemoattractant Protein-1 (MCP-1) Expression, through Upregulating the Early MicroRNA-122 Expression in the Dorsal Root Ganglia. Cells 2020, 9, doi:10.3390/cells9071669.
- Zhao, Y.; Wang, Z.; Zhang, W.; Zhang, L. MicroRNAs play an essential role in autophagy regulation in various disease phenotypes. Biofactors 2019, 45, 844-856, doi:10.1002/biof.1555.
- Xu, J.; Wang, Y.; Tan, X.; Jing, H. MicroRNAs in autophagy and their emerging roles in crosstalk with apoptosis. Autophagy 2012, 8, 873-882, doi:10.4161/auto.19629.
- Li, Y.; Jiang, J.; Liu, W.; Wang, H.; Zhao, L.; Liu, S.; Li, P.; Zhang, S.; Sun, C.; Wu, Y., et al. microRNA-378 promotes autophagy and inhibits apoptosis in skeletal muscle. Proc Natl Acad Sci U S A 2018, 115, E10849-E10858, doi:10.1073/pnas.1803377115.
- Wang, Y.; Zhao, H.; Guo, M.; Fei, D.; Zhang, L.; Xing, M. Targeting the miR-122/PKM2 autophagy axis relieves arsenic stress. J Hazard Mater 2020, 383, 121217, doi:10.1016/j.jhazmat.2019.121217.

Round 2
Reviewer 1 Report
The authors have updated the Western blot examples and provided sufficient explanation. The higher resolution images for Figs 3 and 4 now provide good detail.
Reviewer 2 Report
Article is acceptable.